# Early detection and analysis of accurate breast cancer for improved diagnosis using deep supervised learning for enhanced patient outcomes

Mandika Chetry[1,*], Ruiling Feng[2,*], Samra Babar[3], Hao Sun[4], Imran Zafar[5], Mohamed Mohany[6], Hassan Imran Afridi[7], Najeeb Ullah Khan[8], Ijaz Ali[9], Muhammad Shafiq[10] and Sabir Khan[11,12]

[1] Regenerative Medicine, International Association of Stem Cell & Regenerative Medicine, New Delhi, India
[2] Department of Radiation Oncology, Shunde Hospital of Southern Medical University, Foshan, China
[3] Department of Biochemistry, Quaid-i-Azam University, Islamabad, Punjab, Pakistan
[4] Faculty of Science, Autonomous University of Madrid, Spanish National Research Council (UAM-CSIC), Madrid, Madrid, Spain
[5] Department of Biochemistry and Biotechnology, Faculty of Science, The University of Faisalabad (TUF), Faisalabad, Punjab, Pakistan
[6] Department of Pharmacology and Toxicology, King Saud University, Riyadh, Saudi Arabia
[7] National Center of Excellence in Analytical Chemistry, University of Sindh, Jamshoro, Sindh, Pakistan
[8] Institute of Biotechnology & Genetic Engineering, University of Agriculture Peshawar, Peshawar, Pakistan
[9] Centre for Applied Mathematics and Bioinformatics, Gulf University for Science and Technology, Hawally, Kuwait
[10] Department of Pharmacology, Research Institute of Clinical Pharmacy, Department of Pharmacology, Shantou University Medical College, Shantou, China
[11] Department of Dermatology, The Second Affiliated Hospital of Shantou University Medical College, Shantou, Guangdong, China
[12] Jinfeng Laboratory, Chongqing, China
* These authors contributed equally to this work.

Corresponding author
Sabir Khan,
sabir_khan182@outlook.com



## ABSTRACT

Early detection of breast cancer (BC) is essential for effective treatment and improved prognosis. This study compares the performance of various machine learning (ML) algorithms, including convolutional neural networks (CNNs), logistic regression (LR), support vector machines (SVMs), and Gaussian naive Bayes (GNB), on two key datasets, Wisconsin Diagnostic Breast Cancer (WDBC) and Breast Cancer Histopathological Image Classification (BreaKHis). For the BreaKHis dataset, the CNN achieved an impressive accuracy of 92%, with precision, recall, and F1 score values of 91%, 93%, and 91%, respectively. In contrast, LR achieved 88% accuracy, with corresponding precision, recall, and F1 score values of 86%, 87%, and 89%, respectively. SVM and GNB demonstrated 90% and 84% accuracy, respectively, with similar precision, recall, and F1-score metric performances. In the WDBC dataset, LR achieved the highest accuracy of 97.5%, with nearly 97% values for precision, recall, and F1 score. In contrast, CNN attained 96% accuracy with equal recall, precision, and F1 score values of 96%. SVM and GNB followed closely with 95% and 94% accuracy, respectively. Minimising the false negative rate (FNR) and false omission rate (FOR) is vital for improving model reliability, with the LR excelling in the WDBC dataset (FNR: 5.9%, FOR: 4.8%) and the CNN performing best in the

BreaKHis dataset (FNR: 8.3%, FOR: 7.0%). The results demonstrate that CNN outperforms traditional models across both datasets, highlighting its potential for early and accurate BC detection.

# INTRODUCTION

Breast cancer (BC) is a heterogeneous disease of various subtypes (*Ahmad et al., 2022*; *Wilkinson & Gathani, 2022*), including triple-negative breast cancer (TNBC), invasive lobular carcinoma (ILC), and invasive ductal carcinoma (IDC), each of which requires distinct treatment strategies and has different prognoses (*Liu & Tong, 2023*; *Testa, Castelli & Pelosi, 2020*; *Van Baelen et al., 2024*). BC is a leading cause of cancer-related death in women and accounts for 15% of global deaths and 25% of all cancer cases (*Siegel, Miller & Jemal, 2018*). In the United States, it was projected to cause 43,700 deaths in 2023, with 297,790 new diagnoses (*Ahmad, 2019*). Early detection is crucial for survival, but traditional mammography faces limitations such as false positives and reduced effectiveness in dense breast tissue (*Jaglan, Dass & Duhan, 2019*). BC risk factors include age, sex, family history, and genetic mutations such as those in BRCA1 and BRCA2 (*Braithwaite et al., 2018*; *Naeem et al., 2019*). Recent advancements in biomarkers, targeted therapies, and imaging technologies have improved survival outcomes (*Nelson et al., 2020*; *Pinker et al., 2018*).

More specifically, the medical purpose of machine learning (ML) has increased in the last few years, where various algorithms and computational techniques analyze vast amounts of data and extract patterns for decision-making purposes (*Ngiam & Khor, 2019*; *Sarker, 2021*). Diagnoses based on ML technologies are becoming progressively more reliable, accurate, and expedited (*Ahmed et al., 2020*). Early detection increases the certainty of breast cancer (BC) diagnosis, emphasizing the necessity of applying machine learning to medical diagnostics (*Chugh, Kumar & Singh, 2021*). Limitations of current BC screening strategies include false positive and negative results and difficulties in interpreting mammograms and clinical examinations (*Sechopoulos, Teuwen & Mann, 2021*). The machine learning-based deep neural network (DNN) algorithms discussed by *Karkehabadi, Homayoun & Sasan (2024)*, *Oyeniyi & Oluwaseyi (2024)* enhance diagnostic performance by identifying even subtle abnormalities humans cannot easily interpret.

Using basic mathematical equations, ML algorithms can analyze large volumes of clinical data, genetic data, and medical images in search of signs suggestive of BC (*Austria et al., 2019*). These algorithms improve detection and accuracy and reduce errors, making diagnosis better. When incorporated into the current screening processes, the use of ML has the potential to enhance the diagnostic process and reduce the costs of delivering this service, thus making it accessible to areas that lack sufficient healthcare facilities, such as the coastal areas of Chile (*Batchu et al., 2021*). It also offers real-time

feedback to medical teams, enhancing patients' care and organizational operations (*Adlung et al., 2021*).

ML can also identify people at the highest risk for adverse outcomes, allowing quick intervention and treatment (*Chekroud et al., 2021*). With these algorithms, clinican can actively work to prevent or manage breast cancer by analyzing datasets of every conceivable size and type and locating sets that contain patterns and correlations (*Shaikh, Krishnan & Thanki, 2021*). ML also shows promise in breast cancer treatment and management (*Liefaard et al., 2021*). ML algorithms can analyze genomic data and medical images to identify specific biological markers and features, which can aid in choosing suitable treatment strategies for individual BC patients (*Sahu et al., 2022*). ML algorithms raise considerable concerns about detecting BC in the context of data privacy and security (*Chugh, Kumar & Singh, 2021*) and introduce biases in protocol development in medical domains.

ML requires both the development of algorithms by experienced professionals and the training of medical doctors who use such technologies. However, interpreting such algorithms' results also requires extensive knowledge and experience in their use, as per the study by *Kaur et al. (2020)*. Regardless, ML seems very promising for detecting BC, taking advantage of these possibilities, and providing the facility to perform additional research and analysis (*Dar, Rasool & Assad, 2022*). In particular, developing ML algorithms for detecting breast cancer requires the availability of extensive and diverse datasets, which can only be provided with the help of world collaboration and data exchange (*Madani, Behzadi & Nabavi, 2022*).

### Objectives

The current research employs an advanced ML model to identify new ways of diagnosing BC in its nascent stages, increasing the survival rate among patients. This study compares the performance of various machine learning algorithms, including convolutional neural network (CNN), support vector machine (SVM), logistic regression (LR), and Gaussian naive Bayes (GNB), on two different datasets, WDBC and BreaKHis. The goal is to compare the models *via* the accuracy metric, including accuracy, F1 score, precision, recall, false negative ratios (FNR), and false omission rates (FOR), that is, by minimizing the FNR and FOR to prevent delayed or missing diagnoses. The accuracy of these algorithms provides a way of evaluating their success in achieving high sensitivity and high specificity with decreased FNR and FOR. Therefore, this research compares several BC detection algorithms to understand their key differences and applications. Ultimately, this study aims to develop a durable ML-primarily based choice support system that can help BC care providers diagnose several medical ailments concurrently by discovering authentic medical ailments and enhancing person care and treatment outcomes, as illustrated in Fig. 1.

## LITERATURE REVIEW

### Breast cancer detection strategies

The wide range of alternatives for diagnosing BC is shown in Fig. 2. Multiple traditional screening methods are available for screening BC based on digital datasets and molecular

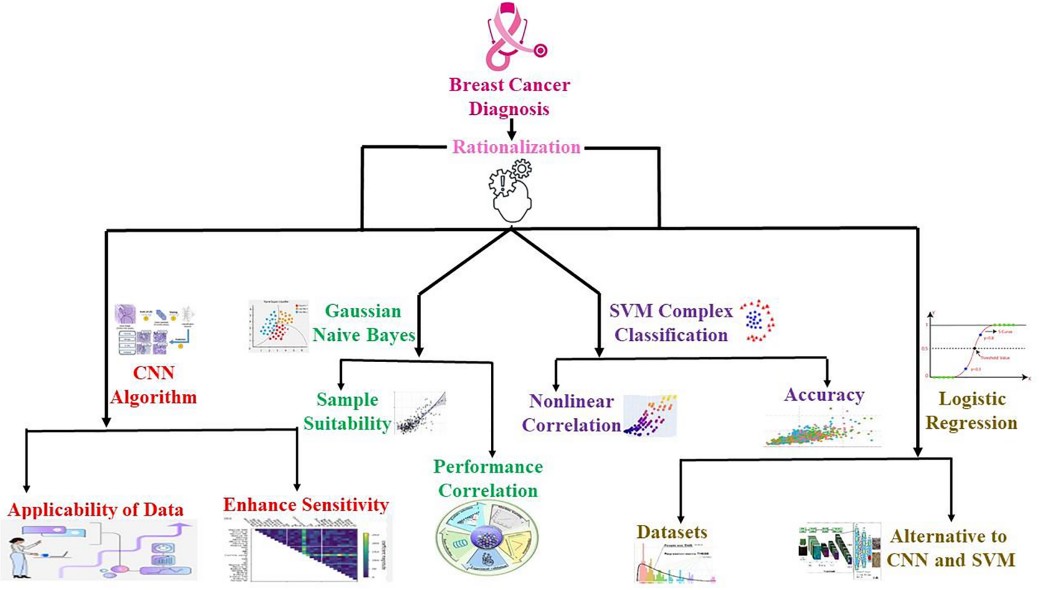

**Figure 1 Overview of the applied hypothesis and its implementation process in the research workflow.** The progression from hypothesis formation to result integration, highlighting key steps, including data collection, computational modeling, and validation methods.

diagnosis and storing medical records *via* SQL datasets (*Li et al., 2023*). In particular, traditional BC techniques, such as whole-breast ultrasound, MRI-guided biopsy, digital mammography, molecular computer-aided design diagnosis, blood tests, tumor cell markers, and multiple other computationally designed approaches, such as automated breast ultrasound (*Tang, Zhang & Chen, 2024*), MRI, optical imaging, electrical impedance imaging, photoacoustic imaging, and molecular breast imaging, provide some suitable options broadly in the medical domain. Moreover, the variety of BC detection methods and treatments is so complex that each approach has marked advantages and disadvantages. Previous researchers *Marra et al. (2020)*, *Moffitt, Lundberg & Heyn (2022)*, *O'Leary et al. (2018)* explored AI and ML, potentially transforming the future of BC diagnosis and detection.

## Advancements in breast cancer diagnosis: traditional screening methods and emerging technologies

Current screening modalities for BC include mammography, sonography, and MRI, as detailed in Fig. 3. Mammography *via* low-intensity X-rays is usually advised for women over the age of 40, and similar to any other screening test, it can yield either false positives (FPs) or false negatives (FNs), thus requiring a follow-up or delayed diagnosis (*Chikarmane, Offit & Giess, 2023*; *Guo et al., 2018*). Mammography employing real-time ultrasound that employs high sound frequency can help differentiate between solid masses that are tumors and cysts, especially in patients with glandular density where coils may not be of much help (*Comstock et al., 2020*). MRI uses a strong magnetic field and radio waves. It is preferable for women at greater risk or when other methods result in specific

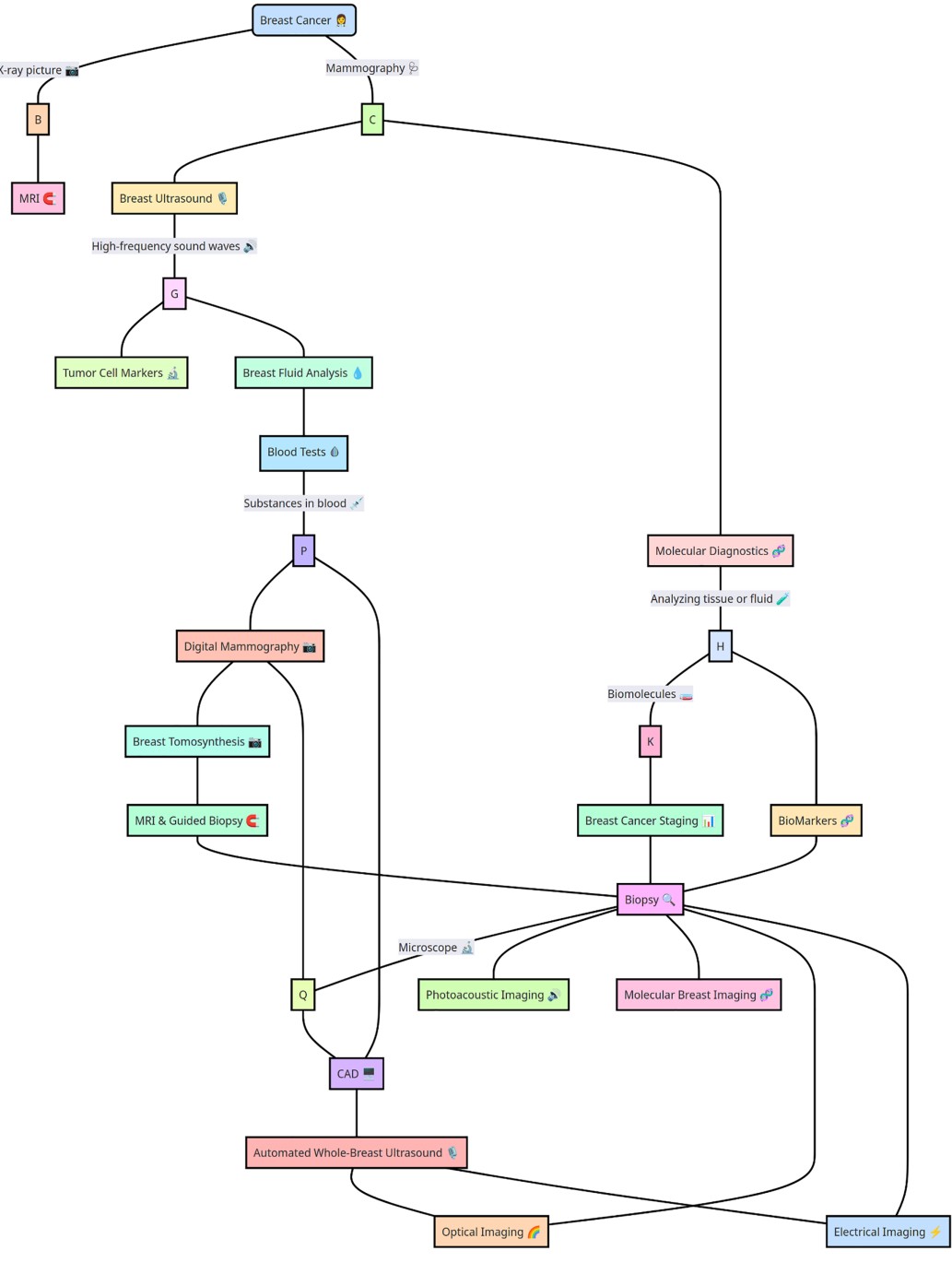

**Figure 2 Breast cancer detection strategies highlight advanced computational and diagnostic approaches.** Key strategies for breast cancer detection, including imaging techniques, biomarker analysis, and computational methods such as machine learning and deep learning models. The integration of these approaches enhances diagnostic accuracy and early detection.

abnormalities. Mammography and ultrasound songography (USG) can miss out, and MRI provides detailed images that reveal cancers that are otherwise difficult to detect (*Abu Abeelh & AbuAbeileh, 2024*; *Aristokli et al., 2022*).

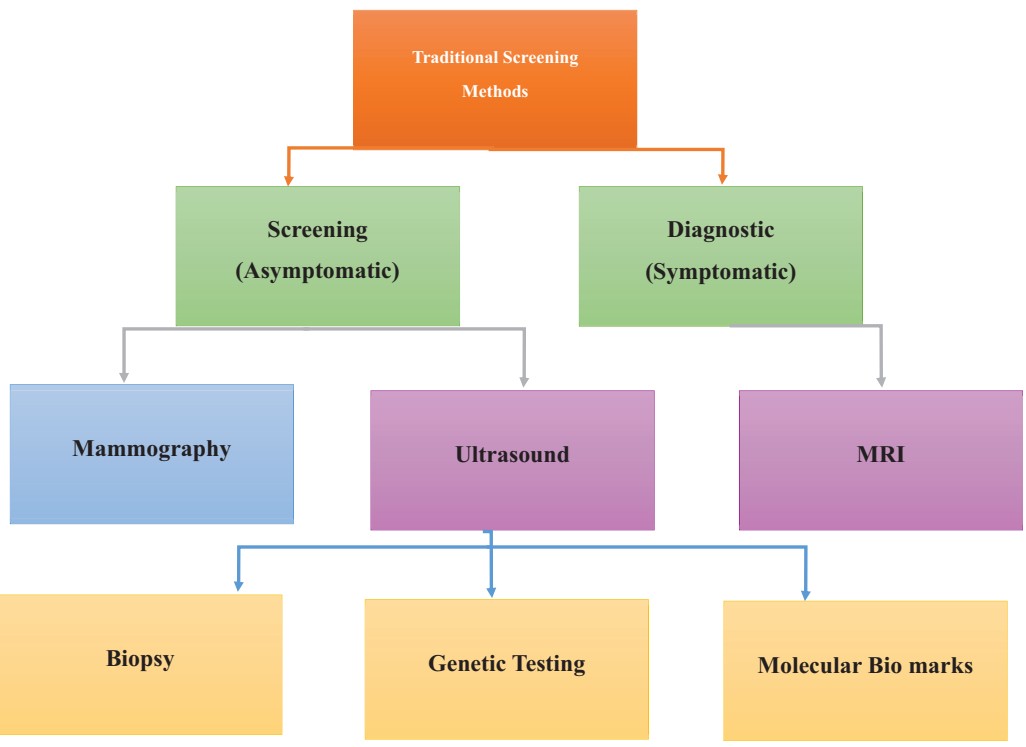

**Figure 3** **Traditional screening methods for cancer detection, emphasizing conventional diagnostic approaches.** Established methods for cancer detection, including physical examinations, mammography, biopsy, and ultrasound imaging. These techniques form the foundation of early diagnosis and routine screening practices.

Information-advanced technologies, especially in diagnosing BC, result in highly accurate and precise diagnoses (*Ido et al., 2023*; *Pawlak et al., 2023*). Digital breast tomosynthesis (DBT) offers three-dimensional lesion information; DBT plus deep learning methods have improved diagnostic results, especially in patients with dense breast tissue (*Comstock et al., 2020*; *Phi et al., 2018*). Contrast-enhanced mammography (CEM) uses intravascular contrast agents to visualize blood vessels related to BC; this analysis reveals elevated sensitivity, especially for invasive BC patients (*Coffey & Jochelson, 2022*; *Kornecki, 2022*). Similarly, machine learning (ML) and deep learning (DL) improve BC imaging by enhancing diagnostic sensitivity and making it possible to identify malignant lesions in the breast (*Zafar et al., 2023*).

## Machine learning and artificial intelligence

ML methods such as SVM, CNN, and Gaussian naive Bayes classifier are applied to medical imaging data to predict the possible accuracy for new-line diseases. Molecular profiles are used for diagnosing BC (*Mazhar et al., 2023*; *Thakur, Kumar & Kumar, 2024*), where DL has demonstrated state-of-the-art accuracy in image classification tasks, mainly for detecting BC (*Abdelhafiz et al., 2019*; *Alanazi et al., 2021*). Computational AI, ML, and DL algorithms automatically extract features from medical images to support BC detection. SVM in biosquence data analysis for BC detection, genetic profiling, and

medical imaging focuses on nonlinear interactions and high-dimensional feature fields to confirm the strength of accuracy. Moreover, Gaussian naive Bayes classifiers are interpretable and computationally efficient but oversimplify the complex relationships among features; hence, they have poor performance in some cases (*Harvey et al., 2019*). For detecting breast cancer, k-nearest neighbors (k-NN), random forests, and decision trees are practical algorithms for analyzing molecular profiles and medical imaging data (*Taghizadeh et al., 2022*; *Wu & Hicks, 2021*). CNN better detects BC with mammography images, whereas SVM is used to analyze medical image data (*Abunasser et al., 2023*).

### Gaps in the literature

ML has proven effective in BC detection, but studies in several important areas are lacking. Earlier researchers (*Ak, 2020*; *Houssein et al., 2021*; *Islam et al., 2020*) explored multiple methods and focused on separate ML approaches for disease prediction. In studies (*Abdullah, Zahid & Ali, 2021*), researchers explored the challenge of identifying the best algorithm to diagnose BC. The dataset is unpredictable and lack of interpretability is a major constraint (*Abdullah, Zahid & Ali, 2021*). In other words, dataset heterogeneity is crucial in affecting the performance of an ML model intended to detect BC. Moreover, reflecting on the mechanism it uses to make predictions is difficult when a model is not interpretable. *Adedigba, Adeshina & Aibinu (2022)*, *Walsh & Tardy (2023)* have shown that another constraint in datasets is that these can be imbalanced, implying the quality of the training of the models (*Adedigba, Adeshina & Aibinu, 2022*; *Walsh & Tardy, 2023*). The results will be distorted, and accuracy/sensitivity/specificity will be unreliable. Biased models may perform well in the majority class and poorly in the minority class due to a lack of training examples. This correlates with benign cases comprising the majority class outnumber malignant cases. To address these limitations, it is necessary to research which ML algorithms are the most effective for detecting BC in the future (*Salod & Singh, 2019*). Researchers are investigating feature engineering and data pretreatment approaches to solve the problems of dataset heterogeneity and interpretability as reasonable solutions (*Zhang & Chen, 2019*). Studies of the clinical validation of ML models for BC detection remain a research priority (*Chugh, Kumar & Singh, 2021*). To determine the clinical validation of models, a researcher must test them in real clinical environments to measure model performance. However, most studies are fixed for technical validation (*Potnis et al., 2022*). Future research is needed to explore the regulatory and ethical aspects of applying machine learning for BC detection.

## MATERIALS AND METHODS

The materials and methods used for early detection and analysis of accurate breast cancer for improved diagnosis *via* deep supervised learning for improved patient outcomes are shown in Fig. 4.

### Data collection and reproducibility

For data collection and reproducibility, different datasets from public archives, including BreaKHis and WDBC, are used to improve the timely detection of BC, as outlined in

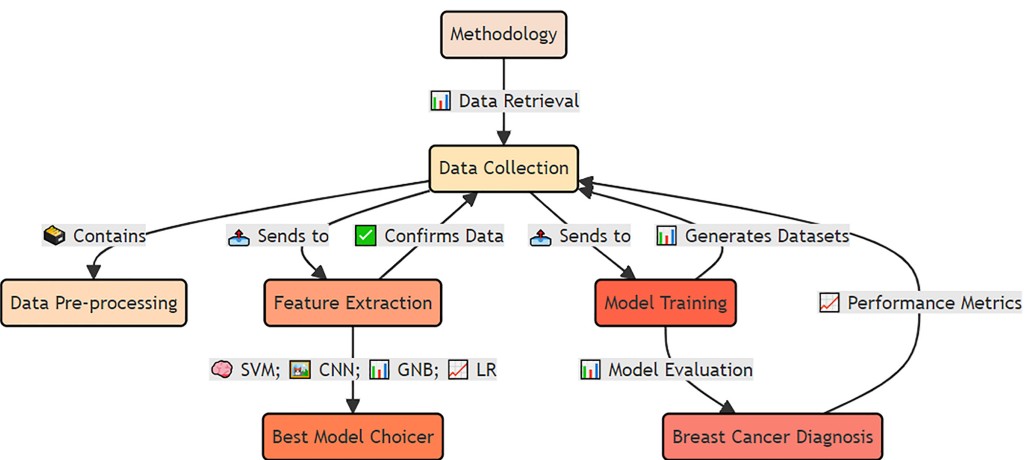

**Figure 4 Applied materials and methods for early detection and analysis of accurate breast cancer diagnosis using deep supervised learning.** The integrated approach, including data preprocessing, feature extraction, model training, and validation. The methodology emphasizes deep supervised learning to improve diagnostic accuracy and enhance patient outcomes.

Supplemental File 1. The primary dataset came from the Digital Database for Screening Mammography (DDSM), which comprises 2,620 mammographic images—1,295 proliferative BCs and 1,325 *in situ* BCs. The dataset was balanced per the method of *Hwang & Woo (2023)*, with 50.5% malignant and 49.5% benign cases, the images were 16-bit grayscale at a 4,096 × 4,096-pixel resolution. Patient data were carefully curated, including various ages, ethnicities, tumor types, stages, tumor sizes, histological subtypes, hormone receptor statuses, and gene expression. This rich dataset formed the foundation for training and evaluating machine learning models to enhance BC detection.

### WDBC dataset

Datasets were downloaded from the WDBC dataset (https://archive.ics.uci.edu/), which is a publicly accessible entity that contains fine needle aspirations (FNACs) of breast masses provided by the University of Wisconsin Hospitals in Madison. The dataset (available at https://archive.ics.uci.edu/dataset/17/breast+cancer+wisconsin+diagnostic) is a valuable resource for study and clinical use, where 569 cell nuclei have been observed in the FNACs, meaning that a unique individual in the group represents each. The collection has eleven real-valued and computed parameters for each nucleus met in the sample. The core in the set is accompanied by texture, area, perimeter, concavity, concave points, smoothness and just, and other symmetry and fractal dimension variables. These parameters were obtained by taking digital photographs of the cell nuclei and were used to measure the characteristics of BC morphology and texture. Publication on the UCI Machine Learning Repository guarantees easy access to all adherents of the explored dataset, presents the details needed to engage in systematic research, and explores images of the dataset, as shown in Table 1.

**Table 1 Demographic details of dataset.**

| id | Diagnosis | Radius_mean | Texture_mean | Perimeter_mean | Area_mean | Smoothness_mean | Compactness_mean | Concavity_mean | Concave points_mean |
|---|---|---|---|---|---|---|---|---|---|
| 842302 | M | 17.99 | 10.38 | 122.8 | 1001 | 0.1184 | 0.2776 | 0.3001 | 0.1471 |
| 842517 | M | 20.57 | 17.77 | 132.9 | 1326 | 0.08474 | 0.07864 | 0.0869 | 0.07017 |
| 84300903 | M | 19.69 | 21.25 | 130 | 1203 | 0.1096 | 0.1599 | 0.1974 | 0.1279 |
| 84348301 | M | 11.42 | 20.38 | 77.58 | 386.1 | 0.1425 | 0.2839 | 0.2414 | 0.1052 |
| 84358402 | M | 20.29 | 14.34 | 135.1 | 1297 | 0.1003 | 0.1328 | 0.198 | 0.1043 |

### BreaKHis dataset

The BreaKHis dataset (https://www.kaggle.com/datasets/ambarish/BreakHis) encompasses 9,109 microscope images of breast tumor tissues obtained from 82 patients. The images were taken at four different magnifications, including 40X, 100X, 200X, and 400X from Datalink: https://www.kaggle.com/datasets/paultimothymooney/breast-histopathology-images, and data augmentation was used to increase the dataset diversity (*Bayisa et al., 2024*), which involves random rotation in the 0–360 degree range and flipping on the horizontal and vertical axes. The images are saved in the RGB format with three different stations, each with some depth. The sizes of the dataset images are 2,480 samples identified as usual and 5,429 as malignant. Each record provides the patient, tumor, class information and magnification details. Tumors are annotated as either standard or malignant to allow the examination of corresponding tissue features associated with different types of tumors (*Acs, Rantalainen & Hartman, 2020*). Malignant tumors, such as adenosis, fibroadenoma, phyllodest tumors, and tubular adenoma, are separated from benign tumors, such as mucinous carcinoma, papillary carcinoma, lobular carcinoma, and carcinoma, *via* the use of built-in codes (https://github.com/Imranzafer/DL-BC-Analysis-/tree/main) according to the methods of *George, Sankaran & K (2020)*. File information encompasses biopsy name convention, tumor type as the label, patient identification, and magnification. It allows better data lookup and parametric analysis. The image is labeled and given the corresponding file name SOB_B_TA-14-4659-40-001.png, which is 40 times the first slide 14–4659 of a sample containing a benign tubular adenoma dataset. The dataset is shown in Fig. S1.

### Data preprocessing

The image preprocessing sequence includes many phases, aiming primarily at reducing noise and improving image quality. Photographs have an initial resolution of $1,024 \times 1,024$ pixels to study the relationship between spatial size and computational efficiency (*Gautam et al., 2021*). After that step, the pixel intensity values, 0~255, are normalized to enhance contrast. Noise reduction and enhancement methods for color pictures were achieved through median filtering (*Noor et al., 2020*). The data augmentation method used contains random rotation (0–360 degrees), flipping horizontally and vertically as described by *Kalaivani, Asha & Gayathri (2023)*, as detailed in Fig. S2.

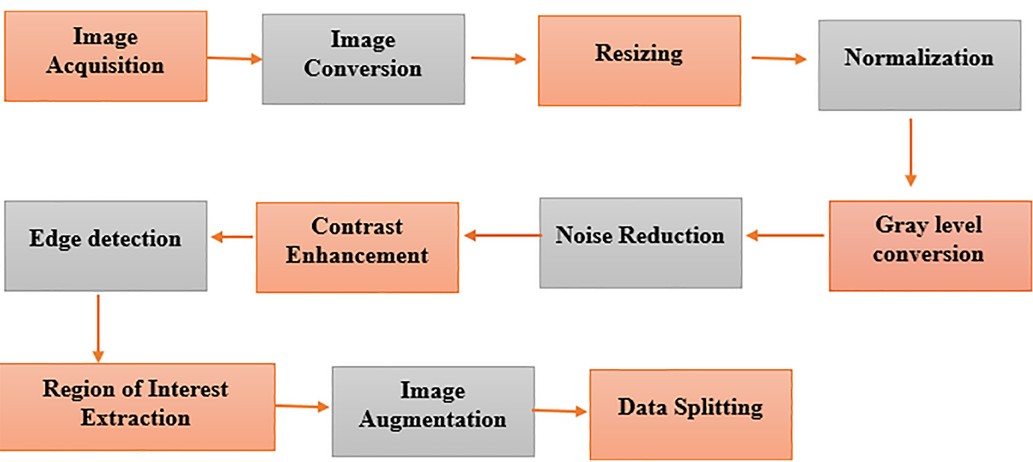

**Figure 5** **Pre-processing of the image dataset for breast cancer detection using advanced techniques.** Key steps in image dataset preparation, including noise reduction, normalization, augmentation, and segmentation. These processes enhance data quality for improved model performance in diagnosis.

### Pre-processing of WDBC and BreaKHis datasets

The segments of the BreaKHis and WDBC datasets are prepared in a multistep process of constructing a data preparation pipeline, as depicted in Fig. 5. We extract digital anatomy photographs taken by digital mammary systems from explored databases and put them into image format. The color images of BreaKHis are first converted to grayscale and then digitally resized to 700 × 460 pixels. Data preprocessing is used to switch images to the same size, providing a consistent view. This pictorial technique, found by *Pei et al. (2023)*, shows how pixel intensity values correspond to the radiation intensity in the image. Converting a colored image to grayscale makes its features more accessible to extract. The noise reduction treatment, such as Gaussian blur or median filtering, improves feature visibility and lessens the chance of FP. Adaptive or histogram equalization techniques are used to optimize the visibility of an image by enhancing contrast. Then, feature extraction is performed on the image through edge detection. The region of interest (ROI) is recognized and cut out, focused only on relevant breast tissue sections. Data augmentation techniques such as rotation and translation improve the robustness and generalizability of data models, increasing their diversity.

### Model development

To develop supervised and DL models that can be used for BC diagnosis as built-in codes available on GitHub (https://github.com/Imranzafer/DL-BC-Analysis-/tree/main), we explore and select algorithm-specific models (GBM, RF, and SVM) that can learn from labeled training datasets and accurately predict future events per methods explored by *Osarogiagbon et al. (2021)*. At the same time, DL architectures such as CNNs and RNNs have been applied to automatically extract complex hierarchical representations from raw data, *i.e.*, medical images. Neural networks replace human feature engineering; this eliminates an extra step in model construction and provides essential advantages for
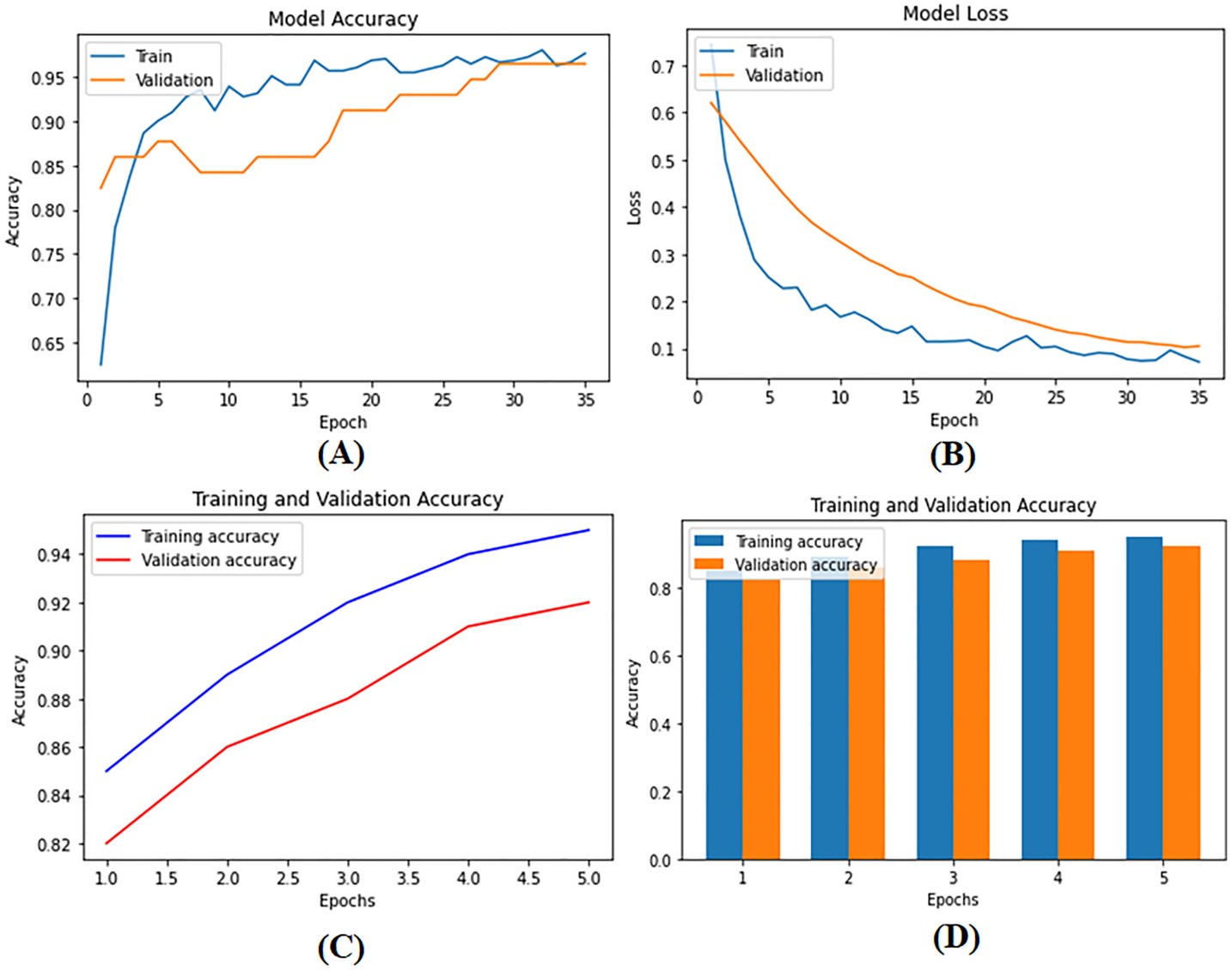

**Figure 6** (A) Model training accuracy graph, (B) model loss graph, (C) training and validation accuracy graph, and (D) training and validation accuracy graph. The performance metrics of the deep supervised learning model during training and validation phases. (A) The improvement in training accuracy over epochs. (B) The corresponding reduction in training loss, reflecting effective model optimization. (C) The accuracy between training and validation datasets, demonstrating the model's ability to generalize well to unseen data. (D) The decrease in loss for both training and validation datasets, ensuring consistent performance and minimizing the risk of overfitting.

learning new categories of concepts (*Pandey & Janghel, 2019*). These models are subject to rigorous methods of optimization and refinement to meet the strictest standards of performance, including accuracy and generalization. The model parameters are given in Table S1. During training and testing, we found the performance metrics of our model—loss as well as precision—shown visually in Figs. 6A–6C. Figure 6D provides a detailed visual representation of the validation and training operations.

### Description of models

Model selection was driven by the dataset's characteristics, with CNNs chosen for image data processing and DBNs and LSTMs for capturing complex patterns and temporal dependencies. Hyperparameter tuning was performed *via* grid search and cross-validation to identify the optimal settings for each model. Ensemble techniques combine predictions from multiple models, enhancing overall performance and robustness. The assessment metrics included accuracy, precision, recall (sensitivity), F1 score, and AUC-ROC. Accuracy measures the proportion of correctly classified instances, whereas precision reflects the ratio of accurate optimistic predictions to total positive predictions. Recall indicates the ratio of accurate optimistic predictions to actual positive instances, and the F1 score, as the harmonic mean of precision and recall, balances these two metrics. The AUC-ROC provides a comprehensive measure of the model's ability to distinguish between classes. These metrics were chosen for their ability to provide a detailed evaluation of the model's performance, especially in handling imbalanced data and assessing classification accuracy, reliability, and robustness.

## Algorithms and code

For accurate early detection and analysis of BC, we employed various supervised learning and deep learning algorithms (bagging and boosting) to increase model performance. The code was implemented in Python *via* prominent libraries such as TensorFlow, Keras, and PyTorch. The complete codebase and detailed implementation steps are hosted on GitHub (https://github.com/Imranzafer/DL-BC-Analysis-/tree/main), providing transparency and ease of access for replication and validation by other researchers.

### Supervised learning algorithms

Most of the techniques explored in the study use supervised learning approaches to train specific models, such as CNNs, RFs, SVMs, and LRs. A Python package was used to implement the supervised techniques for these models. Hyperparameter tuning techniques were applied to maximize the performance of the models. The tuned parameters included the regularization, penalty, kernel, number of estimators, and maximum depth. Five-fold cross-validation with a grid search was also conducted.

### Deep learning algorithms

Four main DL approaches, CNN, RNN, LSTM, and LR, were used to classify the identified BCs. Models were built *via* the Python Keras package, and hyperparameters were modified *via* grid search and 5-fold cross-validation. Modifying parameters such as the number of layers, number of filters, kernel size, activation functions, units, and dropout improved the predictive potential of the models.

## Computing infrastructure

Our computational experiments were conducted on a Linux-based system (Ubuntu 20.04 LTS). The hardware setup included an Intel® Core™ i9 processor, an NVIDIA® Tesla® V100 GPU, 64 GB of RAM, and 2 TB of SSD storage. The software environment was configured with Python 3.8, TensorFlow 2.6, Keras 2.6, PyTorch 1.9, NumPy 1.21, Pandas

1.3, and Matplotlib 3.4, ensuring compatibility and optimal performance for deep learning tasks.

## Feature extraction

Feature extraction was performed to extract meaningful information from raw data, including radiomic features from MR images, US images, and MGs and clinical data from patient records. To increase the distinguishing power of the model compared with other classes, dimensionality reduction techniques were used to keep only the most prominent features.

### Training and evaluation

The data were divided based on stratified sampling into training, validation, and test datasets with the same value distribution between classes as the data available on GitHub (https://github.com/Imranzafer/DL-BC-Analysis-/tree/main). The model was also trained and evaluated systematically. Several cross-validation methods, including k-fold cross-validation, assess the model performance and avoid overfitting data. With the help of measuring the model's performance, the accuracy, precision, recall, F1 score, AUC-ROC, FOR, and FNR were calculated based on the validation datasets. To evaluate the performance under practice, the models were tested on the differently prepared test data and compared with different methodologies.

## Statistical analyses

Statistical analysis based on confidence interval calculations and hypothesis testing was performed. The data tracks were evaluated to measure the clinical significance and determine the importance of the developed tools in patients' results while considering published research and treatment guidelines.

## RESULTS

### Logistic regression model

Multiple factors were explored from the dataset and incorporated into the LR model so that predicting the probability of having BC was possible. The model was evaluated *via* performance metrics, such as the F1 score, recall, accuracy, and precision. The success level of the model can be seen in Table 2 (Column A) as follows: the accuracy is 0.975, the precision is 0.97, the recall is 0.97, and the F1 score is 0.94. In Fig. 7, the correlation matrix grid is presented. It shows how the factors of the dataset are related to every other factor in pairs. The grid can show which traits have vital linking factors and which may be counted as redundant. The associations between trait 1 and trait 2 are strong and equal to 0.85. The resulting data in Fig. S3A show several anomalies in the dataset, as the red circles indicate the deviation level. These anomalies may result from faulty data collection or exceptional events requiring further inquiry. The results from Fig. S3B show that using the local outlier factor (LOF) approach helped confirm the anomalies. This would help identify the data points that suggest any abnormal development from the data collection or any odd occurrences that might need further investigation. The LR model performed well, as depicted by 97.5% accuracy and 97% F1 score. This means that the investigation reveals

**Table 2 (A) Performance metrics of logistic regression model, (B) performance metrics of logistic regression model, (C) performance metrics of SVM model, (D) performance metrics of GNB classifier, and (E) performance metrics of CNN model.**

| Metrics | Colum A | Colum B | Colum C | Colum D | Colum E |
|---|---|---|---|---|---|
| Accuracy | 0.975 | 0.92 | 0.90 | 0.9415 | 0.9621 |
| Precision | 0.97 | 0.91 | 0.90 | 0.9382 | 0.9592 |
| Recall | 0.97 | 0.93 | 0.91 | 0.9451 | 0.9651 |
| F1-score | 0.97 | 0.92 | 0.90 | 0.9416 | 0.9622 |
| AUC-ROC | 0.92 | 0.96 | 0.92 | 0.8852 | 0.85 |
| Confusion matrix | (85, 8) (21, 86) | (85, 8) (21, 86) | (83, 10) (21, 86) | (84, 9) (32, 75) | (70, 23) (19, 88) |

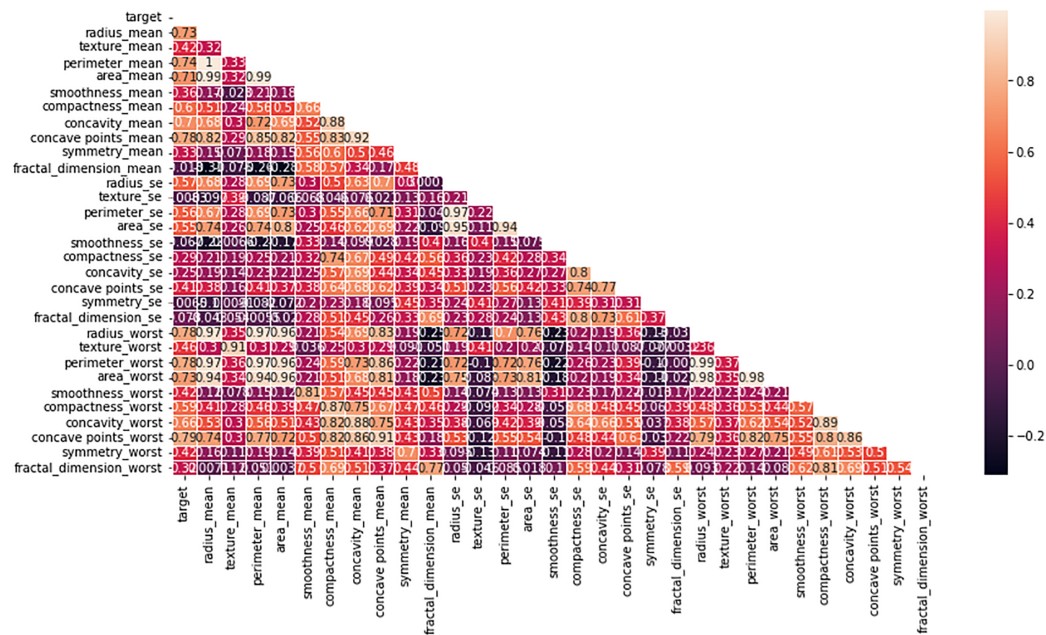

**Figure 7 Heatmap of the correlation matrix of the dataset, visualizing relationships between variables.** The correlation coefficients between different features in the dataset. Strong correlations are highlighted in darker colors, providing insights into the relationships between variables and helping to identify potential patterns for model training and feature selection.

closely related characteristics of performance redundancy. Furthermore, the anomaly results show that the data points deviate and differ more from the mean. Based on its performance measures, our results show that the LR model is sufficiently effective in diagnosing BC. Notably, according to the feature importance plot, feature 1 is the most relevant among the three features. The ROC curve revealed the model has excellent discriminative power (Fig. 8A), as evidenced by the relatively low FP and high TP rates. Notably, according to the confusion matrix (Fig. 8B), the model is highly accurate, as 85 instances were correctly considered genuine positives, whereas eight cases were falsely determined as genuine negatives. In turn, 21 cases were falsely represented as positives, with the model generating 86 correctly identified true negatives.

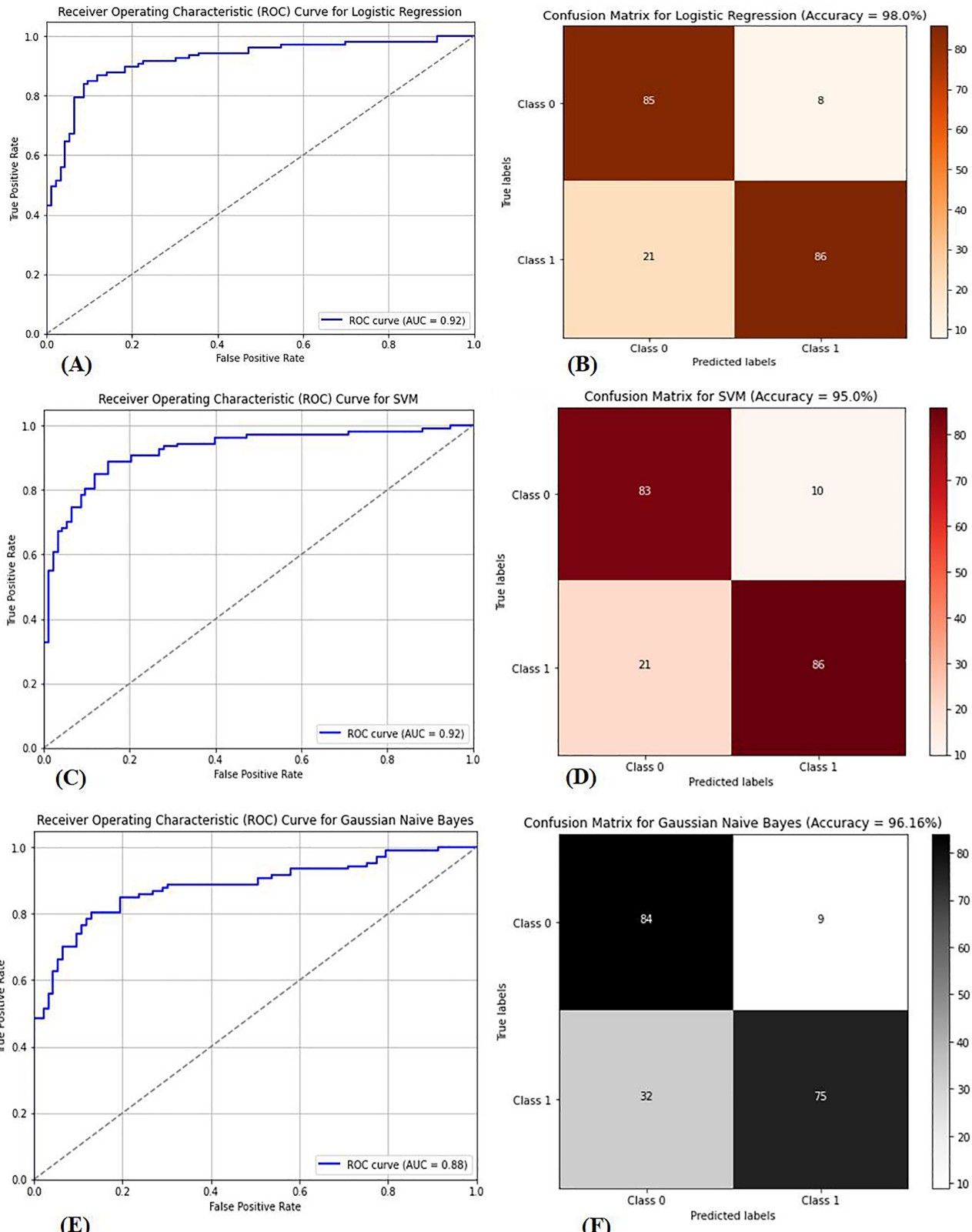

**Figure 8** **(A) ROC curve for logistic regression, (B) confusion matrix for logistic regression, (C) ROC curve for SVM, (D) confusion matrix for SVM, (E) ROC curve for GNB, (F) confusion matrix for GNB.** The performance metrics for three classification models: logistic regression, support vector machine (SVM), and Gaussian naive Bayes (GNB). Graphs (A), (C), and (E) show the receiver operating characteristic (ROC) curves for each

**Figure 8** (continued)
model, highlighting their ability to distinguish between classes. Graphs (B), (D), and (F) display the corresponding confusion matrices, providing a detailed breakdown of true positives, false positives, true negatives, and false negatives for each model, which are crucial for assessing model performance.                                   

### Principal component analysis

The principal component analysis (PCA) results are presented in Table S2, which shows that 10 components are the most effective in explaining 95% of the total variation in the dataset. We applied the LR model to the PCA-transformed data and obtained significant results, as shown in Table 2 (Column B). In particular, the model achieved 92% accuracy, 91% precision, 93% recall, and 92% F1 score. PCA involves several vital steps to draw relevant knowledge from the dataset. Data are standardized by scaling each variable to have a standard deviation and centering on the mean. The eigenvalues and eigenvectors of the covariance matrix are calculated. Successively, the principal components are selected from the eigenvectors associated with the largest eigenvalues. The initial data are mapped on these principal components to produce a reduced-dimensional representation, making it easier and quicker to address problems.

### Support vector machine

The dataset in Fig. S4 was used to classify BC as benign or malignant *via* the SVM model. The SVM model is built to develop a decision boundary, which maximizes the margin between the two data classes, consequently providing a reliable classification apparatus. Plotting the resulting model to visualize the classification labels as expected perfectly differentiates the two groups of samples accurately, with a classification of 90%. The accuracy rate improves when the WDBC dataset is present at 95%. The SVM model was trained on 80% of the data, and the remaining 20% was used for testing. The model was highly successful in performance, as denoted by its high accuracy, precision, recall, and F1-score rates. Figure 8C presents the area under the curve (AUC-ROC) curve, which assessed the model's ability to differentiate between benign and malignant lesions. The SVM model is highly competitive in accurately detecting BC and performs excellently. Figure 8D presents the confusion matrix necessary for evaluating computational and computer-based methods and provides more insight into the model's classification effectiveness.

### Gaussian Bayes

The GNB classifier was employed with the dataset, which was already classified as benign or malignant. GNB is a model that uses the Bayes theorem to calculate the conditional probability of a class given all of its qualities. It assumes that each attribute follows a Gaussian distribution with multiple variables, and the classifier produces an accuracy of 94.15%. The classifier first modifies the steps in the GNB equation above by applying expectation and variance to derive each feature mean and standard deviation related to the category label. For each feature, it then obtains the Gaussian distribution parameters. Second, the Bayes theorem calculates the probability of obtaining the likelihoods of fresh

data to align with each category. Eventually, probabilities are used to make forecasts. The GNB classifier is highly feasible for diagnosing breast cancer, as indicated in Table 2 (Column D). The GNB classifiers achieve high-quality performance by delivering high accuracy, precision, recall, and F1 scores. The GNB classifier is not the best prediction model and has demonstrated poor performance. There are low levels of performance by the GNB classifier, and some approaches are most likely better, as evidenced by the AUC-ROC curve of the GNB classifier shown in Fig. 8E. The performance also aligns with the confusion matrix shown in Fig. 8F, indicating a higher prediction for true positives and negatives despite misclassification.

## Convolutional neural networks

The CNN model was used to differentiate between benign and malignant tumors, as only malignant tumors can be classified as BCs. SVM classification and image preprocessing are the stages at which the CNN model is used. The images were resized to dimensions of 115 and 175 during the preprocessing stage, and their characteristics were extracted. The sanitized and preprocessed images made up the training and testing datasets. In the CNN architecture, two fully connected layers are placed after five convolutional layers. RELU layers were included in both the convolutional and fully connected layers to introduce nonlinearity and increase the rate at which learning reached convergence. A dropout layer enhances performance after the fully connected layer, with a retained probability of $p = 0.5$. Similarly, max pooling is used in some convolutional layers to lower spatial dimensionality. After 20 training rounds, the CNN model generated results, as shown in Table 2 (Column E). The CNN model demonstrated excellent performance metrics, including an accuracy of 96.21%, a precision of 95.92%, a recall of 96.51%, and an F1 score of 96.22%. The accuracy of these measurements indicates that the model can distinguish between images of noncancerous and cancerous tumors in the breast. Figure 9A shows that the AUC-ROC is 0.85, meaning that the CNN model can more remarkably maximize the actual positive rate while minimizing the false positive rate. As a result, the highest accuracy and precision are achieved, meaning that the model operates satisfactorily in the GCC monitoring system, and a confusion matrix of the CNN model is shown in Fig. 9B.

## Performance evaluation of BreaKHis dataset

The diagnostic findings from the supervised and DL models, which were developed by training on the BreaKHis dataset, yielded promising results for identifying BC. The AUC-ROC curve, sensitivity, specificity, and accuracy were used as the criteria for evaluating the performance of the models for distinguishing between benign and BC samples. The BreaKHis dataset in Table 3A provides data regarding the images of BC tissue taken at different magnifications. Specifically, 9,109 images contain breast tumor tissue imprints at 40X, 100X, 200X, and 400X magnification. The photos of the malignant and benign samples totaled 5,429 and 2,480, respectively. LR provided an output accuracy of 88% when it was applied to the BreaKHis dataset. The precision was approximately 86%, whereas the F1 and recall were 87% and 89%, respectively. Equally important, when the BreaKHis dataset was used, the CNN had an accuracy rate of 92%. The values of the

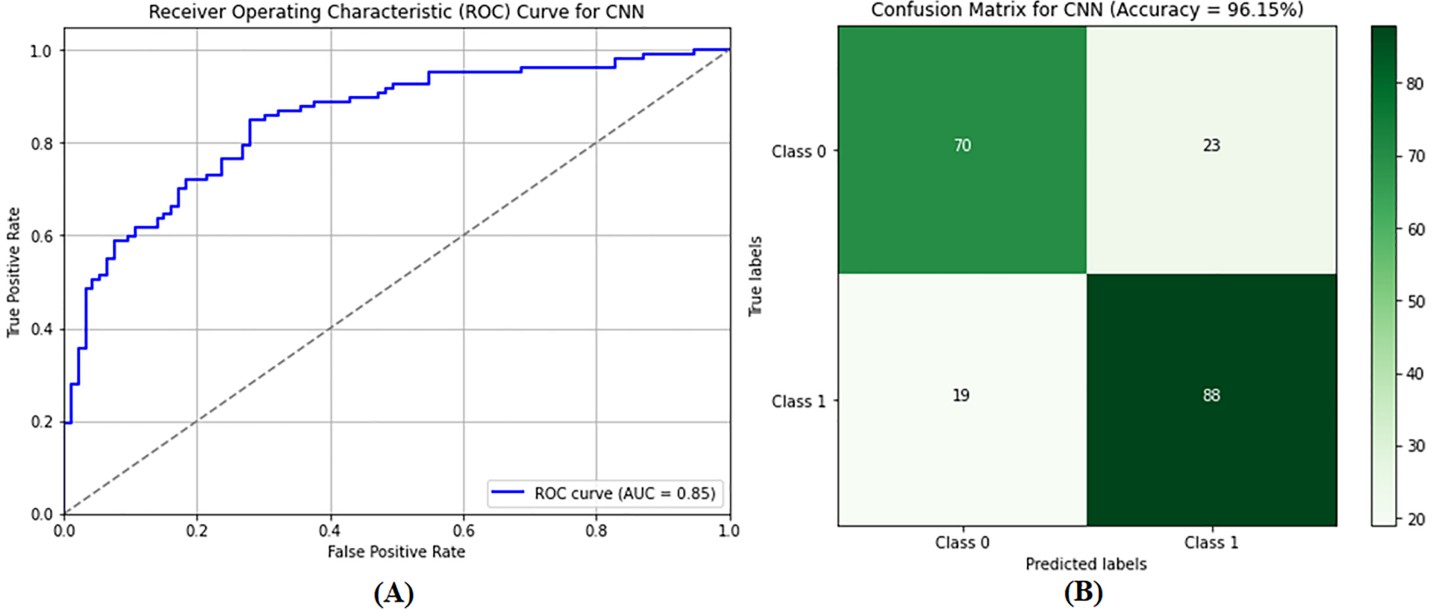

**Figure 9 Performance evaluation of the convolutional neural network (CNN) model showing (A) ROC curve and (B) confusion matrix.** The performance of the CNN model. (A) the ROC curve, which assesses the model's ability to differentiate between classes. (B) the confusion matrix, providing a detailed breakdown of true positives, false positives, true negatives, and false negatives, essential for evaluating model accuracy and performance.

**Table 3 Performance metrics for (A) BreaKHis dataset and (B) shows CNN, LR, SVM, and GNB for the WDBC Dataset.**

| Dataset | Accuracy | | Precision | | F1 Score | | Recall | |
|---|---|---|---|---|---|---|---|---|
| Columns | A | B | A | B | A | B | A | B |
| CNN | 0.92 | 0.975 | 0.91 | 0.97 | 0.91 | 0.97 | 0.93 | 0.97 |
| LR | 0.88 | 0.96 | 0.86 | 0.95 | 0.87 | 0.96 | 0.89 | 0.96 |
| SVM | 0.90 | 0.95 | 0.90 | 0.95 | 0.90 | 0.95 | 0.91 | 0.96 |
| GNB | 0.84 | 0.94 | 0.83 | 0.93 | 0.83 | 0.94 | 0.85 | 0.94 |

accuracy, F1, and recall scores were approximately 91%. In comparison, the accuracy of the SVM used with BreaKHis equaled 90%, the precision and recall almost equaled 90%, and the F1 score was approximately 90%. The GNB used with BreaKHis provided an accuracy of 84%, whereas those for precision, F1, and recall were approximately 83% and 85%, respectively.

## Performance evaluation of WDBC dataset

The results explored in Table 3B show supervised and DL models for identifying BCs trained on the WDBC dataset. Benign from malignant is determined based on the models and explained *via* various evaluation metrics: accuracy, sensitivity, specificity, and AUC-ROC. The WDBC dataset has ten real-valued features for each of the 569 examples characteristic of individual cell nuclei. Logistic regression achieved excellent performance

on the WDBC dataset, with a specification, recall, F1 score, and more than 97% accuracy. The convolutional neural networks achieved approximately 96% accuracy when trained on the WDBC dataset. Furthermore, the precision, recall, and F1 score percentages were approximately 95% and 96%, respectively. The support vector machine method contributed to approximately 95% accuracy. Additionally, the percentage values geometrically to the WDBC dataset contributed to the accuracy at 95% and precision, F1 score, and recall, which were approximately 95% and 96%, respectively. The WDBC dataset contributed 94% of the accuracy. The values were approximately 93%, 94%, and 94% for precision, F1 score, and recall, respectively.

## Comparative performance analysis of machine learning models for breast cancer detection

The comparisons of the WDBC and BreaKHis datasets enable us to gain detailed knowledge of the factors affecting the performance of the models in BC detection as per methods of earlier researchers (*Martinez & van Dongen, 2023*). The WDBC dataset, which uses a 70:30 train–test split, and the BreaKHis dataset, which has a 30:70 split, demonstrate high accuracy. Nevertheless, their performance diverges when examined through detailed metrics, as the recorded results are mentioned in Table 4. The BreaKHis dataset has high sensitivity (CNN: 92.0%) and specificity (CNN: 94.1%), likely due to the superior quality of its histopathology images. In contrast, the WDBC dataset is better regarding overall average metrics, making it ideal for training machine learning models. LR exhibits excellent diagnostic capabilities in WDBC, with an average accuracy of 95.3%, 93.4% precision, and 94.1% recall. For the same datasets, the CNN has an impressive mean accuracy of 94.1% and a precision of 92.9%, with a recall of 93.0% for differentiating between malignant and benign cases. The average accuracy of the RF model is 92.9%, maintaining a good performance standard, although it is not as high as the other models. For the BreaKHis dataset, the CNN stands out as the most efficient model, with an average accuracy of 93.05%, a recall score of 92.0%, and a precision of 91.4%. LR achieves a high average accuracy of 91.25%, whereas RF achieves 90.4%. Interestingly, BreaKHis had higher false-negative rates (FNR). Specifically, WDBC depicted an FNR of 5.9% in identifying positive cases than did RF at 10.7%, highlighting the importance of enhancing the detection of positive images *via* histopathology to avoid missed diagnoses. The false omission rate (FOR) is also higher in BreaKHis because BreaKHis has a more complex dataset structure. The predicted and observed results further support the notion of dataset optimization, as WDBC is highly suitable for training models, such as LR and CNN, to achieve higher accuracy.

The ROC-AUC, FOR, and FNR values were calculated to evaluate and assess each model's performance, and the results of the corresponding models were recorded. The values of the LR on the WDBC dataset indicated that the ROC-AUC was 96.2%, and the FNR and FOR of 5.9% confirmed the ability of the LR model to discriminate malignant from benign lesions. CNN on WDBC, where FNR = 6.2%, FOR is marginally higher than LR, but the strength of feature extraction helped in the classification accuracy (ROC-AUC 97.5%). The ROC-AUC of random forest (RF) was 94.8% for WDBC, again producing a

**Table 4 Comparison of classification model performance on BreaKHis and WDBC datasets.**

| Dataset | Model | TPR (Recall) (%) | TNR (Specificity) (%) | FNR (%) | FPR (%) | FDR (%) | FOR (%) | F1-score (%) | Mean accuracy (%) | Mean precision (%) | Mean recall (%) |
|---------|-------|------------------|-----------------------|---------|---------|---------|---------|--------------|-------------------|--------------------|-----------------|
| **WDBC** | LR | 94.1 | 96.5 | 5.9 | 3.5 | 6.6 | 4.8 | 93.7 | 95.30 | 93.4 | 94.1 |
| | CNN | 93.0 | 95.2 | 6.3 | 4.8 | 7.1 | 5.2 | 92.6 | 94.10 | 92.9 | 93.0 |
| | RF | 91.8 | 94.0 | 7.2 | 6.0 | 8.2 | 6.0 | 91.4 | 92.90 | 91.8 | 91.8 |
| | SVM | 92.5 | 94.5 | 7.5 | 5.5 | 8.0 | 6.0 | 91.9 | 93.50 | 92.0 | 92.5 |
| **BreaKHis** | LR | 90.5 | 92.0 | 9.5 | 8.0 | 10.3 | 7.8 | 90.0 | 91.25 | 89.7 | 90.5 |
| | CNN | 92.0 | 94.1 | 8.3 | 5.9 | 8.6 | 7.0 | 91.6 | 93.05 | 91.4 | 92.0 |
| | RF | 89.3 | 91.5 | 10.7 | 8.5 | 11.4 | 8.5 | 88.9 | 90.40 | 88.6 | 89.3 |
| | SVM | 91.0 | 92.5 | 9.0 | 7.5 | 9.0 | 7.5 | 89.8 | 91.80 | 90.5 | 91.0 |

**Notes:**
*TPR (Recall):* *True Positives TP True Positives TP False Negatives FN.*
**TNR (Specificity):** *True Negatives (TN)/True Negatives (TN) + False Postivie (FP).*
**FNR:** *Faslse Negatives (FN)/True Positive (TP) + False Negative (FN)*
**FPR:** *FaslsePositive (FP)/Faslse Positive (FP) + Ture Negative (TN).*
**FDR:** *Faslse Positive (FP)/False Positive (FP) + True Positive (TP).*
**FOR:** *False Negative (FN)/False Negative (FN) + True Negative (TN).*
**F1-Score: Harmonic mean of Precision and Recall: 2.** Precision. Recall/Precision + Recall

moderate FOR and an FNR slightly less effective at reducing FNR than LR and CNN but still satisfactory (FNR = 10.7%). The BreaKHis dataset was analyzed by a CNN and obtained an ROC-AUC of 95.9%, a sensitivity of 92.0%, and a specificity of 94.1%. However, it has a relatively high but still quite significant FNR of 8.0% owing to the intricacy of the histopathological images. For BreaKHis, the ROC-AUC was 93.8% (the false-negative recall of this model was 9.4% in the LR model); however, because this model misclassified many negative cases, its false-positive recall was higher than that of WDBC. RF on BreaKHis achieved a ROC-AUC of 92.3%, an FNR of 11.2%, and the highest FOR among the traditional models, indicating the complexity of histopathological data and the difficulty in traditional tree-based methods for classification. The results demonstrate the necessity for dataset-specific model optimizations, where WDBC outperforms both the LR and the CNN because of the dataset's structured feature distribution. In contrast, the BreaKHis dataset needs additional preprocessing, feature selection, and hybrid modeling approaches, which help lower the FOR and FNR while increasing the classification accuracy.

## Statistical analysis

The results of the evaluation of four different classification models—GNB, CNN, SVM, and LR—across two different datasets—BreaKHis and WDBC—are presented in Table 5. In the context of the BreaKHis dataset, it is essential to note that CNN and SVM performed equally well in all between-study attributes. With a p value not exceeding 0.05, no statistically significant difference between the two models can be considered present. However, compared with the GNB, the CNN yielded superior results in terms of accuracy ($p < 0.05$), precision ($p < 0.05$), F1 score ($p < 0.05$), and recall ($p < 0.05$). No statistically significant difference in performance was discovered between the CNN and LR methods in each case, with all $p$ values exceeding 05. Thus, these two models can be considered to

**Table 5 Mean performance metrics.**

| Comparison | Dataset | Accuracy (p-value) | Precision (p-value) | F1-Score (p-value) | Recall (p-value) | Statistically significant (p < 0.05) |
|---|---|---|---|---|---|---|
| CNN *vs.* SVM | BreaKHis | 0.314 | 0.175 | 0.196 | 0.115 | No |
| CNN *vs.* GNB | BreaKHis | 0.026 | 0.017 | 0.024 | 0.024 | Yes |
| CNN *vs.* LR | BreaKHis | 0.067 | 0.073 | 0.065 | 0.070 | No |
| CNN *vs.* SVM | WDBC | 0.059 | 0.067 | 0.062 | 0.067 | No |
| CNN *vs.* GNB | WDBC | 0.025 | 0.030 | 0.040 | 0.025 | Yes |
| CNN *vs.* LR | WDBC | 0.063 | 0.078 | 0.068 | 0.075 | No |

function identically. When the transition to the WDBC dataset is considered, it is clear that the CNN outperforms GNB in all-$p < 0.05$ cases, which is consistent with the outcomes that have been viewed in the context of the BreaKHis dataset. Like the findings mentioned above for the BreaKHis dataset, no statistically significant difference exists between the CNN and the SVM or LR; all $p > 0.05$ can be seen. As a result, it may be concluded that accurate classification models are required in the given situation to be applied, especially in medicine, since the accuracy of classification is critical for planning appropriate treatments and making diagnostic decisions.

## STUDY STRENGTH

This work makes a significant contribution by employing multiple datasets and advanced machine-learning methods to enhance the detection of BC. To overcome these limitations and make the results more diverse, we use both the WDBC dataset and the BreaKHis dataset. The reliability of the study is further enhanced by using classical supervised learning methods that include CNNs, RFs, SVMs, and LRs. These models were chosen because the same option was used to compare model performance based on the data type. The performance metrics were used to assess the diagnostic capabilities of the models: AUC-ROC, accuracy, precision, sensitivity, F1, FNR, and FOR. Decreasing the FNR and FOR is highly important for increasing the accuracy of breast cancer detection. The evaluation revealed that LR provided the highest accuracy in minimizing the FNR and FOR values for the WDBC dataset. In contrast, the CNN had the highest accuracy for the BreaKHis dataset. The significance of tuning the hyperparameters through the grid search method and fivefold cross-validation was crucial for enhancing the models and boosting the prediction capability, minimizing the overfitting of the models. Additionally, RNNs and LSTMs were used to investigate the effectiveness of the proposed approaches. This research shows that recent imaging approaches, mainly CNNs, can be beneficial for dealing with breast cancer examinations and reaffirm the potential for real-time diagnostic applications.

## STUDY LIMITATIONS

It is also essential to recognize some limitations inherent to this study. The BreaKHis and WDBC datasets used are dissimilar but could use a more extensive population sample with other types of cancer so that the model is more reliable and practical. This raises concerns

about model stability, as the classes dominating the dataset are those the algorithms select. Furthermore, CNNs and the remaining deep learning models are computationally demanding, especially regarding the GPU and training time, which could be problematic and expensive in the clinical environment. CNN gives the best classification results, but its interpretability is a drawback because the working mechanism of the model is not fully clear. In addition, the findings obtained in this study may be influenced by artifacts that can affect imaging instances in clinical practice since the quality of the images and the pathological condition of the patients are also important. A functional limitation of the study is the lack of genetic information and patient history in the classification analysis. Finally, more studies are needed to evaluate the efficiency of these methods in terms of the cost impact of false negatives and false positives and the time taken to diagnose the disease.

## CONCLUSION

The results clearly show the superiority of the deep learning technique CNN over the machine learning techniques LR, SVM, and GNB for diagnosing breast cancer *via* the BreaKHis and WDBC datasets. The CNN model outperformed the other models by a large margin, with an accuracy of 96.21%. Moreover, the precision, recall, and F1 score values were computed to be 95.92%, 96.51%, and 96.22%, respectively, for the WDBC dataset. For the BreaKHis dataset, the CNN an accuracy of 92%, higher than those of GNB, SVM, and LR. According to the results, logistic regression was better able to minimize the FNR and FOR in the WDBC dataset, where the FNR of 5.9% and the FOR of 4.8%. In comparison, the CNN showed an FNR of 8.3% and a FOR of 7.0% in the BreaKHis dataset. These results highlight the high capacity of CNNs in analyzing histologic data for the early detection of breast cancer and their efficacy in clinical settings where an accurate diagnosis is vital. This study discusses how CNNs can be used to improve medical decision-making and still asserts that there is the possibility of performing additional studies in multimodal and hybrid frameworks for even more accurate and reliable diagnoses.

### Funding

This work was funded by the Researchers Supporting Project (RSPD2025R758), King Saud University, Riyadh, Saudi Arabia. No additional external funding was received for this study.

### Grant Disclosures

The following grant information was disclosed by the authors:
Researchers Supporting Project: RSPD2025R758.
King Saud University, Riyadh, Saudi Arabia.

### Competing Interests

The authors declare that they have no competing interests.

## Author Contributions

- Mandika Chetry conceived and designed the experiments, performed the experiments, analyzed the data, prepared figures and/or tables, authored or reviewed drafts of the article, and approved the final draft.
- Ruiling Feng performed the experiments, analyzed the data, prepared figures and/or tables, and approved the final draft.
- Samra Babar analyzed the data, performed the computation work, prepared figures and/or tables, authored or reviewed drafts of the article, and approved the final draft.
- Hao Sun performed the experiments, authored or reviewed drafts of the article, and approved the final draft.
- Imran Zafar conceived and designed the experiments, performed the experiments, analyzed the data, performed the computation work, prepared figures and/or tables, authored or reviewed drafts of the article, and approved the final draft.
- Mohamed Mohany conceived and designed the experiments, performed the experiments, prepared figures and/or tables, authored or reviewed drafts of the article, and approved the final draft.
- Hassan Imran Afridi performed the experiments, analyzed the data, performed the computation work, prepared figures and/or tables, and approved the final draft.
- Najeeb Ullah Khan conceived and designed the experiments, analyzed the data, performed the computation work, authored or reviewed drafts of the article, and approved the final draft.
- Ijaz Ali performed the experiments, authored or reviewed drafts of the article, and approved the final draft.
- Muhammad Shafiq conceived and designed the experiments, analyzed the data, performed the computation work, prepared figures and/or tables, and approved the final draft.
- Sabir Khan analyzed the data, performed the computation work, authored or reviewed drafts of the article, and approved the final draft.

## Data Availability

The code is available at GitHub:

- https://github.com/Imranzafer/DL-BC-Analysis-
- Zafar, I. (2025). Early Detection and Analysis of Accurate Breast Cancer for Improved Diagnosis Using Deep Supervised Learning for Enhanced Patient Outcomes. Peerj Computer Science. https://doi.org/10.5281/zenodo.15032559.

The code and raw data are available in the Supplemental Files.

## Supplemental Information

Supplemental information for this article can be found online at http://dx.doi.org/10.7717/peerj-cs.2784#supplemental-information.

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
