# Peer review of "Early detection and analysis of accurate breast cancer for improved diagnosis using deep supervised learning for enhanced patient outcomes"

_PeerJ Computer Science, doi:10.7717/peerj-cs.2784_

## Round 0.1 · original submission · Major Revisions

Dear authors,
You are advised to critically respond to all comments point by point when preparing an updated version of the manuscript and while preparing for the rebuttal letter. Please address all comments/suggestions provided by reviewers, and consider how they should be added to the new version of the manuscript.

Kind regards,
PCoelho

Reviewer 1 ·

Basic reporting

1. Compare the suggested method and other baselines more visually.
2. Fig. 1 should be presented in a more scientific style. Now it looks good for a scientific-popular magazine but not for a research journal.
3. Discuss the limitations and potential risks associated with the proposed techniques. Consider computational overhead, potential points of failure, and the trade-offs between accuracy and computational efficiency. I suggest the authors read studies performed by scholars such as s. Jafarzadeh et al., and their groups
4. It is suggested that the conclusion section can also summarize all the findings and attributions, other than the introductions to the proposed methods.
5. Authors should argue their choice of performance evaluation indicators.
The following papers are beneficial to read:
https://doi.org/10.48550/arXiv.2310.00772
https://doi.org/10.1109/ICICIP60808.2024.10477811
https://doi.org/10.1155/2022/5052435
https://doi.org/10.1007/s00500-023-08983-3

Experimental design

As above

Validity of the findings

As above

·

Basic reporting

The authors compared machine learning (ML) algorithms with two datasets, the Wisconsin Diagnostic Breast Cancer and the Breast Cancer Histopathological Image Classification, and found the best results when applying convolutional neural networks (CNN).

* The article is written with technically correct text. Clear and unambiguous, professional English is used in general, but some further improvements are needed: for example, "Hypothetic Results" (line 604) is not an appropriate wording for hypothesis testing.
* The article has a professional article structure and the data and Python scripts were shared and accounted for in the article.

Major comments:
* From a technical perspective, I suggest the authors to add to their references the article of Gonzales-Martinez and van Dongen (2023), due to the reasons described below in Section 2 (experimental design).
* From an epistemological perspective, I suggest the authors to remove the hypothesis framework of the study.

Gonzales-Martinez, R., and van Dongen, D. M. (2023). Deep Learning Algorithms for Early Detection of Breast Cancer: A Comparative Study with Traditional Machine Learning. Informatics in Medicine Unlocked Volume 41, 2023, 101317

https://www.sciencedirect.com/science/article/pii/S2352914823001636

Please be aware that according to PeerJ policies additional references suggested during the peer-review process should only be included if the authors are in agreement that they are relevant and useful.

Minor comments:
* The statement in lines 67-69 ("The most prevalent breast cancer subtypes are triple-negative breast cancer (TNBC), invasive lobular 68 carcinoma (ILC), ductal carcinoma in situ (DCIS), and invasive ductal carcinoma (IDC) have different treatment strategies and prognoses than others") needs to be backed up with a scientific reference.
* The authors talk frequently about differentiating between "benign and malignant BC", a more proper statement is "benign and malignant tumors", since only malignant tumors can be considered breast cancer (BC).

Experimental design

Due to the data imbalance that the authors recognize as a limitation, accuracy is not appropriate measure to compare models, but the F1-score is ok. Also, I advice the authors to include FNR and FOR as performance metrics to compare algorithms. The reasons are described below:

1) FNR measures the proportion of actual positive cases of breast cancer that are incorrectly classified as negative cases, it quantifies the rate of missed positives (Type II errors), and hence a high FNR implies late detection of anomalies.

2) FOR measures the proportion of false negative errors or incorrect omissions in a decision-making process. As FOR captures failures to detect breast cancer, it is also relevant in the comparison of machine learning and deep learning algorithms, because missing the detection of a positive condition of breast cancer can have significant health consequences for cancer patients.

Thus, I suggest the authors to include these metrics in the core evaluation of their proposed models, as in Gonzales-Martinez and van Dongen (2023):

Gonzales-Martinez, R., and van Dongen, D. M. (2023). Deep Learning Algorithms for Early Detection of Breast Cancer: A Comparative Study with Traditional Machine Learning. Informatics in Medicine Unlocked Volume 41, 2023, 101317

https://www.sciencedirect.com/science/article/pii/S2352914823001636

Note that is PeerJ policy that additional references suggested during the peer-review process should only be included if the authors are in agreement that they are relevant and useful.

Additionally, I suggest the authors to remove the hypotheses section, since ML and DL algorithm are inductive by nature, a hypothesis (based on deductive methods) is not needed at all, since the goal of the project is simply to compare algorithms.

Validity of the findings

The validity of the findings and the conclusions linked to the findings should be evaluated on the basis of the lowest FNR and FOR, and not only on the accuracy of the ML and DL algorithms, since, the high level of accuracy found in the paper may be indicative of imbalance problems.

Additional comments

Please be aware that including the citation of the suggested paper is not a requirement for publication, but I will advise to the Editor the publication of the article to consider the publication when FNR and FOR are included in the article as additional metrics to evaluate the performance of the proposed models.

Reviewer 3 ·

Basic reporting

Abstract:
• The abstract succinctly summarizes the research objectives and key findings, underlining various machine-learning models that may be employed for breast cancer detection.
• However, it lacks clarity for the datasets and specific improvements to existing models.
• Briefly mention how this research differs or improves on existing studies in machine learning to detect breast cancer.

Introduction:
• The introduction gives an appropriate background on breast cancer and its challenge of early detection. Besides, it introduces possible solutions through machine learning. However, this literature review feels outdated since most references are from 2018-2021. It's crucial to update the literature with current studies from 2022-2024 to enable a reflection of state-of-the-art developments in machine learning and breast cancer detection.
• Furthermore, the research gap must be further explained and how this study addresses that gap. Hence, I have suggested recent related articles for you to use, cite, and reference.
• Lin, J., Wang, L., Huang, M., Xu, G., & Yang, M. (2024). Metabolic changes induced by heavy metal copper exposure in human ovarian granulosa cells. Ecotoxicology and Environmental Safety, 285, 117078. doi: https://doi.org/10.1016/j.ecoenv.2024.117078
• Chen, L., He, Y., Zhu, J., Zhao, S., Qi, S., Chen, X.,... Xie, T. (2023). The roles and mechanism of m6A RNA methylation regulators in cancer immunity. Biomedicine & Pharmacotherapy, 163, 114839. doi: https://doi.org/10.1016/j.biopha.2023.114839
• Luo, Y., Chen, D., & Xing, X. (2023). Comprehensive Analyses Revealed Eight Immune Related Signatures Correlated With Aberrant Methylations as Prognosis and Diagnosis Biomarkers for Kidney Renal Papillary Cell Carcinoma. Clinical Genitourinary Cancer, 21(5), 537-545. doi: https://doi.org/10.1016/j.clgc.2023.06.011
• Malacara, V. C., Limón, C. G. L., Quintana, O. B., & Macías, G. S. G. (2023). Metastases to pancreas diagnosed by endoscopic ultrasound-guided fine-needle aspiration: A case series and review of imaging and cytologic features. Cytojournal, 20, 18. doi: 10.25259/Cytojournal_55_2022
• Li, Q., You, T., Chen, J., Zhang, Y., & Du, C. (2024). LI-EMRSQL: Linking Information Enhanced Text2SQL Parsing on Complex Electronic Medical Records. IEEE Transactions on Reliability, 73(2), 1280-1290. doi: 10.1109/TR.2023.3336330
• Dai, J., Gao, J., & Dong, H. (2024). Prognostic relevance and validation of ARPC1A in the progression of low-grade glioma. Aging, 16(14), 11162-11184. doi: https://doi.org/10.18632/aging.205952
• Tang, L., Zhang, W., & Chen, L. (2024). Brain Radiotherapy Combined with Targeted Therapy for HER2-Positive Breast Cancer Patients with Brain Metastases. Breast Cancer: Targets and Therapy, 16, 379-392. doi: https://doi.org/10.2147/BCTT.S460856
• Zhou, L., Zhang, Q., Deng, H., Ou, S., Liang, T.,... Zhou, J. (2022). The SNHG1-Centered ceRNA Network Regulates Cell Cycle and Is a Potential Prognostic Biomarker for Hepatocellular Carcinoma. The Tohoku Journal of Experimental Medicine, 258(4), 265-276. doi: 10.1620/tjem.2022.J083
• Chen, S., Zeng, J., Huang, L., Peng, Y., Yan, Z., Zhang, A.,... Xu, D. (2022). RNA adenosine modifications related to prognosis and immune infiltration in osteosarcoma. Journal of Translational Medicine, 20(1), 228. doi: 10.1186/s12967-022-03415-6
• Lastly, briefly mention how this research differs from or improves upon existing studies in machine learning for breast cancer detection.

Objectives:
• The objectives are clear and well-defined, focusing on constructing a machine-learning model to improve early breast cancer diagnosis. The significance and impact of the research could be further emphasized by explicitly mentioning how the study will measure success (e.g., improved accuracy, reduced false positives, etc.).
• It's crucial to explicitly define the performance metrics (accuracy, F1 score, etc.) that will determine the success of the proposed machine learning models.

Literature Review:
• The literature review provides a good overview of traditional and machine learning-based breast cancer detection methods. However, the discussion on existing studies using machine learning feels somewhat general. To ensure the depth and rigor of the study, it's important to include a more detailed and critical analysis of existing studies.
• The authors should critically evaluate the limitations of existing studies and position their work to address these limitations.
• Adding more recent references and identifying research gaps would strengthen the review (use some articles listed in introduction section about).
Hypotheses:
• The hypotheses were clearly stated, each regarding how different machine learning models, such as CNN, SVM, Gaussian Naive Bayes, and Logistic Regression, would perform. However, a few assumptions, such as that CNN performs better than the rest, have not been well-justified.
• Extend the discussion behind each hypothesis regarding previous studies, pointing out why CNN or SVM would perform better for specific contexts.
Materials and Methods:
• The materials and methods are well described, including the datasets, preprocessing techniques, model architectures, and discussions of dataset imbalances and model hyperparameters or optimization processes.
• However, more detailed explanations should be given about handling dataset imbalances.
Results:
• The results section provides a comprehensive overview of the performance of different models on the WDBC and BreakHis datasets. However, the presentation of results could be more organized, and the discussion on model performance could be expanded to include more comparisons to existing methods.
• Include a table comparing each model's performance across various metrics (e.g., accuracy, precision, recall, F1 score). The discussion should also expand to explain why specific models performed better than others.

Discussion
• The discussion provides insights into the findings but could be strengthened by a more detailed comparison of this study’s results to those from similar research. The authors briefly mention the implications for clinical practice but do not explore these in-depth.
• Expand the discussion on how this research contributes to the field of machine learning for breast cancer detection and its potential implications for clinical practice. Provide a more in-depth comparison of the results with other recent studies.

Study Limitations
• The limitation section is well-written, pointing out a relevant problem: most datasets are imbalanced, and the deep learning models are computationally expensive. It could, however, mention other possible limitations that include using more diverse datasets and the applicability of such models in a natural world setting.
• The datasets used may not avoid possible limitations; these need to be discussed considering general geographic and demographic limitations and their likely impact on generalizing the results.

Conclusion
• The conclusion summarizes the essential findings and highlights the CNN models' superior performance. However, it lacks a discussion of future research directions and practical recommendations for deploying these models in clinical settings.
• Add recommendations for future research, particularly on integrating machine learning models into clinical workflows. Additionally, suggest how the models can be validated in real-world clinical environments.

References
• The references section is comprehensive but lacks recent citations. Most references are from 2018-2021, with very few from 2022 or later.
• Suggested Improvement: Update the references to include more recent studies, particularly from 2022-2024, to ensure the literature review is up-to-date.

General Comments
• The manuscript is generally well-written but contains minor grammatical errors and awkward phrasing. A professional editing service could improve its flow and clarity.
• The figures and tables are informative, but some could benefit from more precise labeling and detailed explanations in the captions.

Experimental design

No comment

Validity of the findings

No comment

Additional comments

No comment

---

## Round 0.2 · Minor Revisions

Dear authors,
Thanks a lot for your efforts to improve the manuscript.
Nevertheless, some concerns are still remaining that need to be addressed.
Like before, you are advised to critically respond to the remaining comments point by point when preparing a new version of the manuscript and while preparing for the rebuttal letter.

Kind regards,
PCoelho

**Language Note:** The review process has identified that the English language must be improved. PeerJ can provide language editing services - please contact us at [email protected] for pricing (be sure to provide your manuscript number and title). Alternatively, you should make your own arrangements to improve the language quality and provide details in your response letter. – PeerJ Staff

Reviewer 1 ·

Basic reporting

According to the response letter, the paper has been revised, and the current version of the manuscript is acceptable for publication.

Experimental design

According to the response letter, the paper has been revised, and the current version of the manuscript is acceptable for publication.

Validity of the findings

According to the response letter, the paper has been revised, and the current version of the manuscript is acceptable for publication.

Additional comments

According to the response letter, the paper has been revised, and the current version of the manuscript is acceptable for publication.

·

Basic reporting

I suggest the authors to pay a professional Editor to improve the use of English language in the article, as currently the use of English language is not the best and this is affecting the quality of the article.

Experimental design

Article content is within Aims and Scope of the journal and article type and the methods have improved due to the use of FOR and FNR.

Validity of the findings

I advice the authors to discuss the validity of each of the models/findings not only based on ROC-AUC scores, but also in terms of the values of the FOR and the FNR

Additional comments

The citation of the article of Gonzales Martinez & van Dongen (2023) is not correct, currently it is cited as Martinez & van Dongen 2023, but it should be Gonzales Martinez & van Dongen (2023):

Also in the references, the right citation is:

Gonzales Martinez R, and van Dongen DM. 2023. Deep learning algorithms for the early detection of breast cancer: A comparative study with traditional machine learning. Informatics in Medicine Unlocked 41:101317.

---

## Round 0.3 · accepted · Accept

Dear authors, we are pleased to verify that you meet the reviewer's valuable feedback to improve your research.

Thank you for considering PeerJ Computer Science and submitting your work.

Kind regards
PCoelho

·

Basic reporting

The English languages has improved, but despite the authors claiming to carefully corrected the in-text citation to Gonzales Martinez & van Dongen (2023) and updated the reference section with the correct citation format, issues still persist, please see below.

Experimental design

* In the main text the paper (line 457) the paper is cited as "(Gonzales Martinez R 2023)", it should be
"(Gonzales Martinez R and van Dongen, 2023).

Validity of the findings

And in the references, the paper is cited as:

Gonzales Martinez R vDJIiMU. 2023. Deep learning algorithms for the early detection of breast
637 cancer: A comparative study with traditional machine learning. 41:101317

But it should be:

Gonzales Martinez R, and van Dongen DM. 2023. Deep learning algorithms for the early detection of breast cancer: A comparative study with traditional machine learning. Informatics in Medicine Unlocked 41:101317.